# Future Pharmacists’ Opinions on the Facilitation of Self-Care with Over-the-Counter Products and Whether This Should Remain a Core Role

**DOI:** 10.3390/pharmacy9030132

**Published:** 2021-07-31

**Authors:** Lezley-Anne Hanna, Alana Murphy, Maurice Hall, Rebecca Craig

**Affiliations:** Medical Biology Centre, School of Pharmacy, Queen’s University Belfast, 97 Lisburn Road, Belfast BT9 7BL, UK; amurphy77@qub.ac.uk (A.M.); m.hall@qub.ac.uk (M.H.); rebecca.craig@qub.ac.uk (R.C.)

**Keywords:** over-the-counter medicines, pharmacy, questionnaire, undergraduate

## Abstract

Background: The aim was to investigate pharmacy students’ views on the role of the pharmacist in facilitating self-care with over-the-counter (OTC) medicines, particularly in light of new roles, and establish personal practice. Methods: Final year pharmacy students at Queen’s University Belfast were invited to participate. Data were collected via a pre-piloted questionnaire, distributed at a compulsory class (only non-identifiable data were requested). Descriptive statistics were performed, and non-parametric tests were employed for inferential statistical analysis (responses by gender). Results: The response rate was 87.6% (78/89); 34.6% (27/78) males and 65.4% (51/78) females. Over a third [34.6% (27/78)] reported using OTC medicines about once a month. All appreciated the importance of an evidence-based approach to optimize patient care. Most [(96.2% (75/78)] deemed OTC consultations should remain a fundamental responsibility of pharmacists and 69.2% (54/78) thought OTC consultations have the potential to be as complex as independent pharmacist prescribing. Females felt more confident recommending OTC emergency contraception than males (*p* = 0.002 for levonorgestrel and *p* = 0.011 for ulipristal acetate). Many [61.5% (48/78)] considered more medicines should not be deregulated from prescription-only status. Conclusions: Data from this single institution suggests that enabling self-medication is an important part of practice but there were confidence issues around deregulations.

## 1. Introduction

The facilitation of self-care is important for pharmacists worldwide, as recently demonstrated by the International Pharmaceutical Federation (FIP) 2021 development goals program which includes “Shaping the future of self-care through pharmacy” (International Pharmaceutical Federation, 2021) [1]. Indeed, from a United Kingdom (UK) perspective, there has been a fundamental shift in healthcare policies driven by national self-care initiatives [2]. These initiatives have aimed to empower the public to take responsibility for managing their own health and reduce prescribing costs and demand on the National Health Service (NHS) [2]. It is timely to promote self-care and self-medication given health services may be at breaking point due to the COVID-19 pandemic. In tandem, there has been a commitment from the UK government to improve public access to medicines through the escalation of prescription-only medicine (POM) deregulations to pharmacy-only (P) or general sale list (GSL) status [3]. These latter two legal categories of medicines are collectively known as over-the-counter (OTC) medicines.

The scope of POM reclassifications has expanded beyond those products which are indicated for minor ailments to give pharmacists the capacity to contribute to the management of longer-term clinical conditions [4]. Having the ability to manage a diverse range of conditions capitalizes on the clinical expertise of pharmacists. In addition to treating minor ailments, UK-based pharmacists can now offer such medicines as tamsulosin for benign prostate hyperplasia, sildenafil for erectile dysfunction, ulipristal acetate for emergency contraception, orlistat for weight management and tranexamic acid for heavy menstrual bleeding. In the UK in 2021, there were two public consultations about the deregulation of the contraceptive pill desogestrel [5,6], which received support from a key UK organization, the Faculty of Sexual and Reproductive Healthcare [7] and subsequently two brands (Hana^®^ and Lovima^®^) are soon to be available in UK pharmacies to purchase OTC. Furthermore, a recent survey conducted by the Proprietary Association of Great Britain (PAGB) among over 2000 members of the UK public found that 69% of respondents were now more likely to consider self-care as a primary treatment option and 31% were more likely to visit a pharmacy for initial advice than was the case before the pandemic [8]. Additionally, with the accelerating global issue of antimicrobial resistance and the emergence of initiatives such as the Global Respiratory Infection Partnership, pharmacists will have an integral role in advocating the use of self-care measures to minimize unnecessary use of antibiotics [9].

OTC medicines and consultations are not without risk or safety concerns. The potential for misuse or abuse with such medicines has resulted in more stringent restrictions on sales of ephedrine and pseudoephedrine-containing products [10], opioid-containing products [11], and laxatives [12]. Additionally, within the last six years, potential cardiac effects have resulted in the reclassification of OTC domperidone [13] and diclofenac [14] back to POM status in the UK. Other concerns include potential drug interactions. For example, OTC products containing opioids or sedating antihistamines can interact with other medicines that also have central nervous system depressive effects, potentially affecting the person’s ability to perform skilled tasks. Oral OTC ibuprofen has a substantial interaction profile, including with lithium (increases the concentration), methotrexate (increases the risk of toxicity and gastrointestinal bleeding), and warfarin and systemic corticosteroids (increased bleeding risk). Moreover, in the UK in 2017, OTC miconazole oral gel became contraindicated in patients taking warfarin due to bleeding events (some of which were fatal) [15]. OTC medicine use could also delay medical diagnosis or mask symptoms of a more serious underlying health condition. Self-care is reliant on improving the health literacy of the public and equipping patients with the knowledge to make informed decisions [1], therefore future pharmacists and pharmacists must be appropriately trained in this area. In tandem with this, the practice of evidence-based medicine has become ingrained in the healthcare system and is recognized by the FIP and World Health Organization (WHO) as the expected standard for all patient interactions [16] but research conducted with pharmacists revealed that evidence of effectiveness and adopting an evidence-based approach is a secondary consideration during OTC consultations [17,18,19,20,21]. Lastly, the time available to focus on a comprehensive pharmacist-led OTC consultation may be compromised due to additional new roles such as influenza and COVID-19 vaccine services.

Previous studies in the last ten years involving students have investigated perceptions and practice of self-medication in Bangladesh (500 medical and pharmacy students [22] and 250 pharmacy students [23]), Ethiopia (380 medical and pharmacy students [24]) Iran (170 medical and pharmacy students [25]), Jordan (1317 medical and pharmacy students [26]), Pakistan (300 medical students [27]) and Sri Lanka (700 undergraduate students [28]). Other researchers in Nepal examined students’ knowledge and perceptions about self-medication practices (620 students [29]). Additionally, a qualitative study (semi-structured interviews with 10 pharmacy students) was conducted in Australia to ascertain decision making about over-the-counter medicines [30]. Other studies have investigated self-care education provision across ten Canadian pharmacy schools [31] and approaches used to teach pharmacy students about OTC medicines in Iran [32] and across the globe [33]. Limited research has been conducted to determine UK-based pharmacy students’ self-medication practice or to glean pharmacy students’ opinions on a breadth of related areas (evidence-based practice and decision-making, the importance of pharmacist-led OTC consultations given other new roles, confidence with deregulations and restrictions imposed by manufacturers) [34].

Two authors of this paper conducted previous work with 154 UK pharmacy students [34] over five years ago, however, this was before (i) pertinent deregulations such as ulipristal acetate (emergency contraception with no specific age limit) and sildenafil (for erectile dysfunction) (ii) reclassifications of diclofenac and domperidone back to POM status and the deepening opioid crisis, (iii) new additional roles for pharmacists such as vaccination services and (iv) the launch of new General Pharmaceutical Council (GPhC) standards for pharmacy education which will result in an autonomous independent prescribing qualification for pharmacists at the point of registration [35].

It is anticipated that the findings of this current research will be valuable to shape necessary curriculum changes due to the new pharmacy education standards. Additionally, the authors hope to see where improvements are required to enhance students’ competence and confidence in OTC consultations (which have shifted beyond minor ailments) to better prepare them for their future careers as prescribers of both self-limiting and more complex conditions. The overarching aim of the research was to investigate Queen’s University of Belfast (QUB) final year pharmacy students’ views on OTC medicines, including their place within the pharmacist’s expanding role.

The objectives were to:establish students’ use and views on OTC medicines and factors influencing their decision-making processes (in a personal and professional context)determine the perceived importance of OTC consultations given the evolving role of the pharmacist and the additional new rolesgather opinions on current deregulations (including their level of confidence recommending these) and views on future deregulationsascertain whether gender affected responses

To provide readers with more background information about the QUB pharmacy degree program up to the point of the study, the following subject areas have been taught: organic and bioorganic chemistry, medicinal substances, drug discover and design, drug delivery for large and small molecules, quantitative drug analysis, physiology, clinical pharmacology and therapeutics (including the management of self-treatable conditions and complex patients), evidence-based healthcare, law and ethics (including evidence-based decision models and prescribing principles), key skills such as written and oral communication, physical examination skills, critical appraisal, reflection, decision-making and diagnosis, teamwork, numeracy, statistics, equality, diversity and inclusion. In terms of learning environments, professional aspects of pharmacy practice are taught in a mock pharmacy largely via simulation (including role play scenarios), through work-based learning, and numerous supporting resources are available via the virtual learning environment [in addition to those used in practice and readily available via the Internet such as National Institute for Health and Care Excellence (NICE) Clinical Knowledge Summaries (CKS)]. With regard to the teaching about self-treatable conditions and OTC consultations specifically, the clinical areas covered are eye & ear, gastrointestinal system, nervous system and musculoskeletal injuries, pediatrics, respiratory system, skin conditions and infections, travel health, and women’s and men’s health. Students attend weekly 3.5 h classes for nine weeks.

## 2. Materials and Methods

The subjects were students enrolled in their final year of the QUB Master of Pharmacy (MPharm) degree program (excluding the research student and co-author, Alana Murphy, n = 89). The MPharm degree is a four year degree with two 15 week-semesters per year. As previously mentioned in the Introduction, final year students have greatest insight out of all year groups and therefore are best placed to provide reasoned views on the subject area. Moreover, a substantial part of the questionnaire relates to deregulated prescription-only medicines and the students are only taught about these when they start final year. Additionally, the research is more relevant to them than the other levels as they are soon to graduate and enter the workplace where they will be expected to provide reliable answers to queries about over-the-counter medicine.

Data were collected by means of a self-completed paper-based questionnaire which was developed with reference to relevant publications [22,23,24,25,26,27,28,29,30] including the previous study [34] and the electronic medicines compendium (eMC) [36]. To maximize the response rate [37], the questionnaire was quite short and divided into four discrete sections to ensure it was easy to follow. Additionally, it consisted of primarily closed-ended questions with a 5-point Likert scale (strongly agree, agree, neither agree nor disagree, disagree and strongly disagree) for responses. Several open-ended questions were also used. Section A (11 items) concerned students’ use of OTC medicines and factors affecting product selection for personal use. Section B (12 items) explored students’ views on OTC consultations and factors which may influence their decision-making process when making recommendations to patients in practice. It also aimed to glean views on the place of OTC consultations and medicines within the pharmacist’s evolving role. Section C (26 items) related to deregulated medicines, future reclassifications, and whether OTC product licensing was too restrictive. Section D (1 item) aimed to collect non-identifiable demographic information from the participants about their gender. Other parameters such as ethnicity and age could have uniquely identified students.

A Participant Information Sheet and Questionnaire Cover Sheet were also prepared in line with ethics committee requirements. The questionnaire was piloted with ten postgraduate students or postdoctoral staff in QUB School of Pharmacy. As a result of the pilot, no changes were made to content and structure of the questionnaire and an estimated completion time of 10 min was ascertained.

Following ethical approval from the QUB Faculty of Medicine, Health and Life Sciences Research Ethics Committee (MHLS 20_135; 30 October 2020), an invitation to participate in the voluntary study was emailed to the intended study population This also contained the Participant Information Sheet as an attachment. The following week in November 2020, the questionnaire was manually distributed to students at a compulsory class.

In terms of data analysis, coded responses from the completed questionnaires were entered into Microsoft Excel^®^ version Office 365 (Microsoft Corporation, Redmond, WA, USA). Statistical analysis predominantly took the form of descriptive statistics such as frequencies and percentages. Inferential statistical analysis was also conducted for ascertaining differences in responses by gender. The Mann–Whitney U-Test was applied on ordinal data and Pearson’s Chi-squared and Fisher’s Exact Tests were applied on categorical data, with significance set at *p* < 0.05. Responses to open-ended questions were analyzed using thematic analysis [38].

## 3. Results

The response rate for questionnaire completion of this voluntary study was 87.6% (78/89). In terms of gender of the 78 respondents, 27 reported being male (34.6%) and 51 reported being female (65.4%), 0 selected the ‘prefer not to say’ option and 0 selected the ‘other, please state’ option. Out of the 78 respondents, 77 completed the questionnaire in its entirety and 1 omitted a few statements. The number of respondents who completed each statement is specified throughout. If readers wish to peruse the raw data from the individual questionnaires, this is provided within a spreadsheet in the Appendix A and is entitled Appendix A.

### 3.1. Personal Use of OTC Medicines and Factors Influencing Product Selection (Section A of the Questionnaire)

Most of the student respondents [98.7% (77/78)] reported self-medication with OTC products with only 1 [1.3% (1/78)] stating that they never use OTC medicines (with the rationale that they “never need any OTC medicines as health is good”). The frequency (and %) of OTC medicines use reported by the 78 respondent users is as follows:7 (9.0%). about once a week27 (34.6%). about once a month19 (24.4%). about once every three months11 (14.1%). about once every six months7 (9.0%). about once a year6 (7.7%). less than once a year1 (1.3%). never

As can be seen from the above information, the most frequently selected response was ‘about once a month’. Females were more likely to select this than males [43.1% (22/51) females versus 18.5% (5/27) males, *p* = 0.001]. The most frequently reported category among male respondents was ‘about once every three months’ [25.9% (7/27)].

Those who used OTC medicines (77, as 1 of the 78 respondents stated ‘never’) were asked to outline which OTC medicines they had used over the preceding six months. Of these 77, 74 provided a response but 3 others did not offer any details. Among the 74 who answered this question, the predominant class of medicines used was analgesics [97.3% (72/74)]; this included paracetamol, ibuprofen, naproxen and aspirin. Additionally, 23.0% (17/74) reported using cough, cold and sore throat remedies, with decongestants being the most cited medicine in this category. Antihistamines for allergies were used by 13.5% (10/74) of respondents while products pertaining to the gastrointestinal system were used by 6.8% (5/74).

The third question of Section A asked the OTC medicine users to reflect on factors that influence their choice when selecting an OTC product for personal use. The findings from this question are presented in Table 1 (with the verbatim statements from the questionnaire). Effectiveness, advice from a pharmacist or counter assistant, and safety were the main considerations. One student respondent selected ‘other’ and wrote “quantity in box with how likely I’ll need again”.

### 3.2. OTC Consultations and Decision-Making in Clinical Practice, and the Place of OTC Medicines in the Evolving Role of the Pharmacist (Section B of the Questionnaire)

The first seven statements in this section concerned safety and evidence of effectiveness in relation to carrying out OTC consultations. Respondents’ views on these statements are illustrated in Figure 1 (the verbatim statements from the questionnaire are provided in the figure). Regarding the eighth statement, females were more likely than males to agree that personal experience using OTC medicines strongly influences the products they would recommend in practice [88.2% (45/51) of females ‘strongly agreed’ or ‘agreed’ compared with 66.7% (18/27) males; *p* = 0.04].

The last four statements related to the place of OTC medicines in the pharmacist’s evolving role. Responses and verbatim questionnaire statements are presented in Table 2.

### 3.3. Deregulation of Medicines from Prescription-Only to Over-the-Counter Status and Manufacturers’ Product License/Marketing Authorization Restrictions (Section C of the Questionnaire)

The first question in this section sought to ascertain pharmacy students’ level of confidence in recommending various deregulated OTC products for the appropriate clinical indication. The verbatim statement was: “I would feel confident recommending this medicine to a patient if the appropriate clinical situation arose” to which respondents could select strongly agree (scored as 5), agree (scored as 4), neither agree nor disagree (scored as 3), disagree (scored as 2) or strongly disagree (scored as 1). Responses corresponding to the 23 OTC products, with wording about the products as provided in the questionnaire, are illustrated in Figure 2. Female respondents were more confident than male respondents about making recommendations of levonorgestrel (*p* = 0.002) and ulipristal acetate (*p* = 0.011) for emergency contraception. Interpolated median scores for confidence recommending levonorgestrel were 4.18 (males) versus 4.77 (females). This was similar for ulipristal with interpolated median scores of 4.15 (males) and 4.65 (females).

Respondents were asked to consider whether more prescription-only medicines should be deregulated to OTC status in future. Most (61.5%, 48/78) selected ‘no’, whilst 38.5% (30/78) selected ‘yes’. Of the 48 students who selected ‘no’, 35 provided justification for doing so. The primary reasons provided were concerns for patient safety including potential for abuse of OTCs (15/35), additional responsibility and workload for pharmacists in relation to consultations and monitoring requirements (13/35) and training factors such as risk of misdiagnosis or drug interactions (4/35). Of the 30 students who selected ‘yes’, 24 provided examples of products which they deemed suitable for future deregulation with the most common being oral contraceptives (6/24), oral and topical antibiotics (6/24) and salbutamol inhalers (4/24).

In relation to product licenses, 42.3% (33/78) of respondents ‘strongly agreed’ or ‘agreed’ that OTC medicine licenses were generally too restrictive, whilst 42.3% (33/78) of students disagreed with this statement. Furthermore, 44.9% (35/78) of respondents ‘strongly agreed’ or ‘agreed’ that the recommended duration of use stipulated by manufacturers of OTC medicines rarely align with the usual time taken for the resolution of self-limiting conditions.

## 4. Discussion

The key factor influencing pharmacy students when selecting OTC medicines for personal use was product effectiveness. In a professional context, it was recognized by all that using an evidence-based approach for OTC consultations would enhance the quality of patient care, however ultimately the focus was on safety rather than evidence of effectiveness. While respondents agreed that OTC consultations and the management of self-limiting conditions should remain a fundamental role of the community pharmacist, and could be as complex as newer roles, there was a lack of confidence around some deregulated products and a reluctance for further deregulations.

Almost all respondents reported self-medicating with OTC products which mirrors the findings of the previous study [34] and other work undertaken in Europe [39]. Identical to the previous study [34], the most selected option for frequency of use was ‘once a month’. Female students reported more regular use of OTC medicines than males, which has been previously reported for adult populations in the UK and European countries [40,41,42]. In this current study, the most frequently reported class of medicine used was analgesics, followed by cough, cold and sore throat products, and products for allergies. These results are broadly similar to the previously UK study which was conducted around the same time of the year [34] and a study conducted in Jordan (1317 medical and pharmacy student participants; 1034 of whom reported self-medication practice) where 79.9% reported using analgesics, 33.8% used antitussives and 18.2% used antihistamines [26]. Pharmacy students in Bangladesh [23] and Ethiopia [24] reported commonly using analgesics. Moreover, the latest PAGB OTC consumer healthcare market review indicated that pain relief represented the largest market share, followed by cough, cold and sore throat products [43]. A high prevalence of cough and cold remedies was anticipated as data collection occurred in November when there is seasonal demand for such products coupled with COVID-19 symptom profile. Almost all students (97.4%) agreed that effectiveness was an important attribute when selecting an OTC product for personal use. Interestingly, despite reporting using cough and cold products as outlined above, a Cochrane review concluded that there is no evidence to either support or refute the effectiveness of cough products in acute cough [44] and another found that multiple doses of nasal decongestants may have a small beneficial effect in adults with the common cold but the clinical relevance is unknown and there is insufficient high quality evidence to enable firm conclusions to be drawn [45]. Effectiveness has also been shown to be the primary determinant in OTC product selection among members of the public (n = 1461) in Northern Ireland (which is part of the UK) [42].

In a professional rather than personal context, all student respondents agreed that adopting an evidence-based approach for OTC consultations would enhance patient care. This is consistent with the findings of the previous study [34] and professional guidance [46] and is encouraging for a healthcare profession that places great importance on practice being underpinned by science. Many students agreed that selling OTC medicines which lack evidence is an ethical dilemma; an issue which has previously been recognized by pharmacists in the UK [17]. Juxtaposed with this, students attached importance to the placebo effect in the treatment of minor ailments. The influence of the placebo effect is widely recognized among healthcare professionals in the literature [17,47]. Despite demonstrating a theoretical appreciation of the benefits of an evidence-based approach, the execution of this in practice may be hindered by factors including pressure from patients, time constraints, and a desire to avoid conflict [17].

In terms of deregulations, there were mixed results for confidence recommending each of the medicines provided. Students were most confident recommending oral non-steroidal anti-inflammatory drugs (NSAIDs), namely ibuprofen and naproxen. This was similar to the findings of the previous study [34] and may emanate from the well-documented safety profiles in an OTC context and the robust evidence of effectiveness for their OTC indications to validate recommendations of these products in practice [48,49,50]. Interestingly, while overall confidence about supplying emergency contraception (levonorgestrel and ulipristal acetate) was high, female participants had greater confidence about this than males. This may be linked to personal use; female participants may have particular interest in these products as they are indicated for females within the age range of the study cohort, therefore they may require knowledge of these products in a both a personal and professional context. Respondents were least confident recommending intranasal triamcinolone acetonide. This is difficult to explain, given their views on other corticosteroid formulations, but may be due to a lack of opportunity to recommend this seasonal product (it is licensed for the symptoms of seasonal allergic rhinitis) and because there are other more established OTC intranasal corticosteroids such as fluticasone propionate and beclometasone dipropionate available for the same indication [36].

Less than half of respondents (38.5%) deemed that more medicines should be deregulated from POM status. A comparable opinion was reported in the previous investigation [34] and since then there has been greater focus in the degree program about the benefits of deregulations to patients and the pharmacy profession [51]. Perhaps these future pharmacists consider that the increasing prevalence of independent pharmacist prescribers and the introduction of Patient Group Directions (PGDs), through which pharmacists can supply prescription-only antibiotics for uncomplicated urinary tract infections under strict instruction [52], mitigates the need for more reclassifications. A considerable proportion of students (61.5%) were opposed to further deregulations, and many justified this standpoint by stating they had concerns about patient safety (because of interactions, misdiagnosis, incomplete patient records, and misuse potential). The provision of more clinical and patient-facing training throughout the degree program may help to alleviate such concerns. Moreover, Community Pharmacy Northern Ireland (CPNI) have put forward proposals for an integrated computer system that would provide access to electronic care records as these are not readily accessible [53], thereby enabling pharmacists to tailor the advice and treatment offered to individual patients and reduce the risk of unforeseen medication interactions.

Like the previous study [34], over 40% respondents in this current study considered that product licensing of OTC medicines is too restrictive. The restrictions imposed by manufacturers can be barriers to OTC consultations. For example, orlistat 60 mg capsules are available to purchase OTC, but patients taking medicines for diabetes, hypertension and hypercholesterolemia must be referred to their GP first [36] despite the strong association between increased weight and these comorbidities. There are many other contraindications, precautions and drug interactions and additional warnings specified in the product literature [36] resulting in the population for whom this medicine is suitable being very narrow. Drug interactions include ciclosporin, oral anticoagulants, antiepileptic medicines, antiretroviral medicines for HIV, levothyroxine, amiodarone, antidepressants, antipsychotics (including lithium) and benzodiazepines [36]. Secondly, a similar proportion also agreed that the duration of use of OTC products outlined by manufacturers does not correspond to the typical time taken for symptom resolution. For example, NICE sets expectations that an acute cough may last three to four weeks [54], yet product literature specifies that patients must contact a doctor if symptoms worsen or do not improve after 7 days [36].

Almost all students agreed that OTC consultations should remain a fundamental role for the pharmacist which demonstrates an appreciation for the pharmacist’s contribution to this area of practice. Vaccine administration and independent prescribing represent some of the more recent responsibilities within the pharmacist’s professional profile [55]. Respondents in the current study deemed that OTC consultations have the potential to be as complex as these newer roles and therefore should have an equally significant place within the pharmacist’s scope of practice. Moreover, most student respondents disagreed that the supervision of P medicine sales should be the responsibility of a pharmacy technician, which further affirms the place of OTC consultations falling under the pharmacist’s remit. However, it should be noted that these students only have a theoretical understanding of independent prescribing and perhaps their views on the complexity of it will change when they undertake the role in practice. The new pharmacy education standards [35] appear to place greater emphasis on clinical skills and prescribing aspects of the pharmacist’s role. The challenge for educators will be to find a way to incorporate the new responsibilities in the degree program whilst retaining the value of other roles such as OTC consultations. It is already known from previous work that the teaching of self-care and OTC medicines to pharmacy students is inconsistent [31,32,33]. For example, when Nakhla and colleagues investigated self-care education in ten Canadian schools of pharmacy, they found that education provision varied extensively. This led them to suggest that strategies to enhance current programs, including core requirements (such as hours and topics), may be useful [31]. Additionally, when Sinopoulou and Rutter examined approaches used to teach about OTC medicines from a global perspective (84 completed questionnaires were received from pharmacy schools in 24 countries), they concluded that there was mainly appropriate teaching about diagnosis and management of symptoms but that teaching methodologies could be reviewed to move beyond lectures, tutorials and workshops. They also warned that as the scope of OTC consultations widens, potentially including diagnostic aids and physical examination skills, there is a risk that students are being taught by educators who are not necessarily qualified to teach such skills [33]. Having an appropriate skill set will also be relevant for independent prescribing teaching.

Regarding strengths and weaknesses of this work, the high response rate attained (87.6%) suggests that non-response bias was not a concern. The timing of questionnaire distribution was opportune in that it coincided with the latter half of the simulated classes about OTC consultations and students had recently completed a lecture series on evidence-based healthcare. Students therefore had a good understanding of all terms used and the topics covered were likely to be of interest. However, the sample size was small, and participants were recruited from one higher education institution only. Therefore, the results may not be representative of pharmacy students enrolled elsewhere. The authors consider that, with minor adaptations (for example to the list of deregulated medicines), the questionnaire-based study could be replicated by other establishments across the globe. This work is being presented within the short Communication category in the hope that it stimulates readers to ask similar questions about education and training for OTC consultations in light of evolving roles and FIP’s development goals for pharmacy-led self-care initiatives.

In terms of implications, educators will be required to prepare students to prescribe both OTC medicines and POMs in their new role as an independent prescriber so it could be useful to use an integrated approach and heed the lessons learnt about barriers to effective OTC consultations from this current study and previous research [4,17,31,32,33,34]. Moreover, experience from the UK and other countries such as, Australia, Canada, New Zealand and the United States of America about pharmacist prescribing [56,57] will be invaluable to inform future practice If a UK pharmacy graduate qualifies as an independent prescriber at the point of registration (licensing), it is possible that many who subsequently work in the community sector will manage similar clinical conditions as they are doing currently. A main difference will be that they will be able to do so more comprehensively rather than referring to another prescriber. For example, several skin conditions such as acne and psoriasis can be managed with various topical OTC products. In the future, other topical and oral POM products could be prescribed by the pharmacist independent prescriber where appropriate. This concept will apply to other conditions such as migraine and dysmenorrhea. Additionally, being able to identify red flags requiring urgent referral to hospital remains unchanged. The underlying principles of evidence-based healthcare and practice being underpinned by science also remain priorities for students to learn how to implement. Pressures and expectations from patients will be equally great in a POM prescribing context and there will be numerous products that future pharmacist prescribers will have to be confident recommending and prescribing. Manufacturers could assist by changing the wording of product literature in some instances. The emphasis on patient safety must remain but advice within the patient information leaflet could better align with best practice guidelines.

## Figures and Tables

**Figure 1 pharmacy-09-00132-f001:**
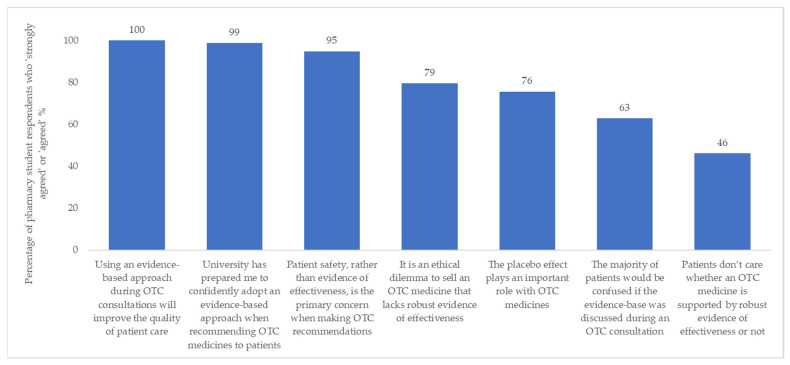
Pharmacy students’ (n = 78) views on the importance of evidence in the context of OTC consultations and decision-making in their capacity as future healthcare professionals.

**Figure 2 pharmacy-09-00132-f002:**
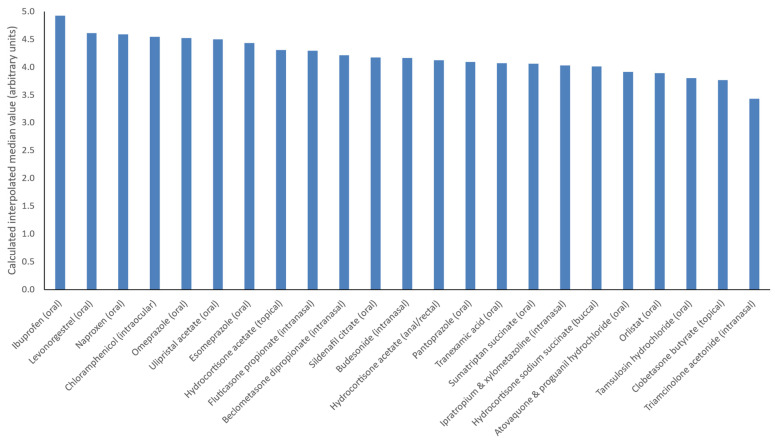
Pharmacy student respondents’ (n = 78) confidence about recommending deregulated OTC products for the appropriate clinical indication. This is presented using the interpolated median of the respondents’ scores for each product and ranked by the highest to lowest interpolated median scores.

**Table 1 pharmacy-09-00132-t001:** Factors that influence pharmacy students (n = 77) when selecting an OTC medicine for personal use, ranked from highest to lowest order of importance as determined from interpolated medians.

	Strongly Agree, n (%)	Agree, n (%)	Neither Agree nor Disagree, n (%)	Disagree, n (%)	Strongly Disagree, n (%)	Interpolated Median (Maximum Possible Score Was 5)
How effective I think the product will be (how well I think it will work)	55 (71.4)	20 (26.0)	1 (1.3)	1 (1.3)	0 (0.0)	4.80
Recommendations or advice provided by a counter assistant or pharmacist	42 (54.5)	30 (39.0)	2 (2.6)	3 (3.9)	0 (0.0)	4.58
How safe I think the product is for me	41 (53.2)	29 (37.7)	4 (5.2)	2 (2.6)	1 (1.3)	4.56
How easy the medicine is to take or use	23 (29.9)	41 (53.2)	10 (13.0)	1 (1.3)	2 (2.6)	4.12
The cost of the medicine	17 (22.1)	44 (57.1)	7 (9.1)	7 (9.1)	2 (2.6)	4.01
Familiarity with the name/brand	8 (10.4)	31 (40.3)	18 (23.4)	11 (14.3)	9 (11.7)	3.52
Product presentation	7 (9.1)	25 (32.5)	14 (18.2)	22 (28.6)	9 (11.7)	3.04
Product advertising	5 (6.5)	12 (15.6)	18 (23.4)	31 (40.3)	11 (14.3)	2.39

**Table 2 pharmacy-09-00132-t002:** Pharmacy students’ (n = 78) views on the place of OTC medicine consultations within the context of the evolving role of the pharmacist.

	Strongly Agree, n (%)	Agree, n (%)	Neither Agree nor Disagree, n (%)	Disagree, n (%)	Strongly Disagree, n (%)	Interpolated Median (Max Possible Score Was 5)
Despite evolving clinical roles, OTC consultations and the management of self-limiting conditions should remain a fundamental role for the community pharmacist	45 (57.7)	30 (38.5)	3 (3.8)	0 (0.0)	0 (0.0)	4.63
OTC consultations have the potential to be as complex as independent prescribing	21 (26.9)	33 (42.3)	12 (15.4)	10 (12.8)	2 (2.6)	3.95
OTC consultations have the potential to be as complex as vaccine administration	22 (28.2)	27 (34.6)	17 (21.8)	10 (12.8)	2 (2.6)	3.87
Supervision of Pharmacy (P) medicine sales should be the responsibility of a pharmacy technician, not a pharmacist	2 (2.6)	8 (10.3)	16 (20.5)	44 (56.4)	8 (10.3)	2.20

## Data Availability

Data is contained within the article or Appendix A. The data presented in this study are available in the Appendix A (i.e., the data from the individual questionnaires is within a spreadsheet within the Appendix A).

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
