# Peer review of "Future Pharmacists’ Opinions on the Facilitation of Self-Care with Over-the-Counter Products and Whether This Should Remain a Core Role"

_pharmacy, 2021, doi:10.3390/pharmacy9030132_

Round 1

Reviewer 1 Report

This was well-written. I do not recall a need to change any word or phrasing. 

Minimal comments attached.

I asked about considering 3 articles that might be of use to the work. In one (CPJ), a huge discrepancy was found in the attention paid to minor ailment education in Canada. So, similarly in your institution, the attention paid to this manifests in how good our students are. 

I also commented on the process of bringing "evidence" into an OTC consult. The process in pharmacies requires a very patient-centered approach, where you just can't get into the weeds on "evidence-based evidence". So, I was surprised to see 2 items on the far right score low. Taking steps not to confuse a patient, and to stay away from all the detailed evidence in my brain, is key to an appropriate consult, often with time restraints.

I picked "average" for scientific soundness, not as a slight, but that it simply meets what we all would expect. All okay. 

Reviewer 2 Report

Overall this is a fine manuscript; it is generally well written, the research itself is well constructed and the authors have been proportionate in their application of findings to the discussion, carefully noting this is a small and limited study.  The authors did a good job of taking what seems to be a relatively straightforward topic and making it interesting.  I have no substantive comments or suggestions to make, other than potentially citing work from other jurisdictions where de-regulation has been more extensive or where pharmacists are more directly involved in prescribing decisions....experience from these jurisdictions may be illuminating for these authors.  Beyond that the study design was reasonable and the analysis and conclusions were straightforward and appropriate - no major concerns from me with respect to advancing to publication.

Reviewer 3 Report

This manuscript addresses an interesting subject as is the knowledge and opinions of pharmacy students about OTC products. Sometimes, we do not give enough emphasis to them when they can have important safety implications.

Therefore, I find the manuscript very interesting and that can give important insights and considerations for the elaboration of study plans in pharmacy.

Some things that I would like to comment:

  • Introduction: In lines, 61-62 the author explain the patient safety and the possible misuse or abuse as problems of OTC products. I think that they should include or make more explicit the possible problems that could be derived from the interactions with other drugs prescribed.
  • Methods:
    • Could the author include more details about the MPharm? How many years/ terms does it last?
    • The version of Microsoft Excel used should be indicated
  • About OTC consultation a decision-making, does the student are aware that exist guidelines that can help in this process?
  • From what I have read in lines 352-57 I deduce that community pharmacists do not have access to the medical records. This is a genera problem in many countries, and I agree with the authors that it is a basic need in order to provide a safe and consistent recommendations, etc. to the patients.
